# When Sardines Disappear: Tracking Common Dolphin, *Delphinus delphis*, Distribution Responses Along the Western Iberian Coast

**DOI:** 10.3390/ani15111552

**Published:** 2025-05-26

**Authors:** Sarah Brouder, Tiago A. Marques, Nuno Oliveira, Pedro Monteiro, Jorge M. S. Gonçalves, Ana Marçalo

**Affiliations:** 1Faculdade de Ciências e Tecnologia, University of Algarve, Campus, 8005-139 Faro, Portugal; 2Centre for Research into Ecological & Environmental Modelling, School of Mathematics and Statistics, University of St Andrews, St Andrews KY16 9LZ, UK; tiago.marques@st-andrews.ac.uk; 3Departamento de Biologia Animal, Centro de Estatística e Aplicações, Faculdade de Ciências da Universidade de Lisboa, 1749-016 Lisboa, Portugal; 4Conservation Department, Sociedade Portuguesa para o Estudo das Aves—SPEA, 1700-031 Lisboa, Portugal; nuno.oliveira@spea.pt; 5Centre of Marine Sciences (CCMAR/CIMAR LA), Campus de Gambelas, University of the Algarve, 8005-139 Faro, Portugal; pmontei@ualg.pt (P.M.); jgoncal@ualg.pt (J.M.S.G.)

**Keywords:** dolphins, marine ecosystem conservation, opportunistic sighting data, predator/prey relationship, purse seine fishery, sardine population dynamics, species distribution modelling

## Abstract

The common dolphin (*Delphinus delphis*) is the most abundant cetacean along the western Iberian Peninsula and faces significant anthropogenic threats, particularly bycatch. This study analyses this species’ distribution along the western Iberian coast using vessel survey data (2005–2020) to identify hotspot areas and seasonal patterns in relation to sardine (*Sardina pilchardus*) abundance. Hotspots were found in central–western and southern coastal areas, overlapping with key fishing ports and sardine juvenile habitats. Notably, between 2013 and 2016, dolphins were observed farther offshore, coinciding with a period of low sardine biomass near the coast. These findings provide essential insights for conservation and the sustainable management of the sardine purse seine fishery, which frequently interacts with this species.

## 1. Introduction

Marine conservation efforts are often based on the knowledge of where species are distributed in space and time [1]. This knowledge can provide key information about priority seasons and areas for protection, therefore informing cetacean conservation and management plans [2,3,4,5]. In marine and coastal management, the use of Species Distribution Models (SDMs) plays a key role in informing Marine Spatial Planning (MSP) initiatives [6,7,8] by providing insight into potential areas of conflict between different users of the maritime space [9,10].

Cetacean SDM can be used to assess where a species’ habitat overlaps with anthropogenic activities, such as fisheries [11], marine traffic ways [12], physical and acoustic disturbance [13,14], seismic activities [15], touristic operators [16], and offshore development [17,18,19]. Understanding how anthropogenic activities overlap and interact with marine mammals’ distribution is fundamental for decision-making regarding the allocation of no-take zones in important migratory routes or breeding and foraging sites [20].

Cetaceans are considered important umbrella species due to their ecological significance and the broad protection their conservation provides to other marine species and habitats [21,22]. As apex predators, they play a crucial role in maintaining ecosystem structure, regulating dynamics, and supporting essential ecological functions [23]. Moreover, they are particularly vulnerable to human activity because of their slow maturity, low reproductive potential, and occurrence in areas with strong human activity [24].

Cetacean conservation is a top priority for marine management plans because their protection can have positive effects on the overall ecosystem health [25]. Furthermore, cetaceans are considered charismatic species and have a strong intrinsic value due to their complexity and diversity [26]. Therefore, accurate abundance and distribution estimates are needed in order to safeguard these flagship species and to address the requirements of the Marine Strategy Framework Directive (MSFD) and the European Union (EU) Habitats Directive [27].

The common dolphin, *Delphinus delphis*, is one of the most abundant cetacean species, with a global distribution [28,29], and is the most abundant cetacean species off the western Iberian coast, with an estimated average population of 80471 individuals [30]. Moreover, studies have shown that the common dolphin is the most frequently stranded species along the Portuguese mainland coast and accounts for the highest number of recorded mortalities associated with fisheries [31,32,33,34,35,36].

Studies have shown that common dolphins favour diets rich in small pelagic fish with high-energy content [33,37,38], changing their diet or energy richness in response to seasonal and yearly fluctuations of prey distribution and abundance [33,37,38,39,40,41,42]. Optimal foraging theory predicts that diet selection is driven by prey availability, catchability, energetic value, and the specific cost of foraging [43], which is the case for predators that lead an energetically expensive lifestyle, such as the common dolphin [38].

The western Iberian waters are known for their high productivity, largely driven by the seasonal upwelling regime [44,45]. This regime sustains the presence of small pelagic fish species whose success depends on favourable environmental conditions [46,47]. However, reported environmental changes, coupled with excessive fishing, have led to declines in the abundance of the European sardine, *Sardine pilchardus* [48,49,50,51].

The sardine stock in western Iberia is part of the southern stock, jointly managed by Portugal and Spain. The sardine stock is mostly targeted by their purse seine fisheries, which are responsible for the bulk of sardine landings [33,52]. Sardine abundance depends on the magnitude of recruitment, which was low from 2004 until recent times [47], leading to a historical decline in biomass. Consequently, a management plan for the southern Iberian sardine stock was implemented by both countries’ administrations in 2012, including significant cuts to landing quotas. This multi-annual management plan, known as the “Sardine Fishery Management Plan (2012–2015)” (henceforth, “Sardine Ban”), established annual catch limits, defined sardine fishing prohibition periods, imposed limits on juvenile catches, and regulated technical measures to reduce bycatch in the Portuguese purse seine fishery (PPSF) [52,53].

The Sardine Ban resulted in a notable recovery in sardine stocks, with a 30% increase in biomass from 2015 to 2016 alone [54]. However, despite some partial increases, stock levels in 2017 were still below the biological limits (Blim) as defined by the International Council for the Exploration of the Sea (ICES) and a long way from economic and ecological stability [54].

Studies have shown a strong interaction between the common dolphin and the activities of the PPSF along the Portuguese mainland coast, where this species is reported to be the most frequently sighted cetacean species within the fishery’s operational zone [32,35,55,56,57]. This strong association between the common dolphin and the PPSF is perhaps unsurprising due to their mutual preferred target, the sardine [31,35,54]. This resource overlap has raised concern, particularly regarding the risk of bycatch, which has been reported in the area [31,32,33,35].

The aim of this study is to provide insights into the spatial and temporal distribution of common dolphins along the western Iberian coast, utilising information from annual non-dedicated vessel surveys conducted between 2005 and 2020. By examining changes in encounter rates and proximity to the coast across different periods, this research investigates the potential influence of the Sardine Ban and fluctuations in sardine biomass on dolphin distribution. Specifically, the study addresses two hypotheses:**Hypothesis** **1:**The Null Hypothesis (H_0_): *There is no statistically significant difference in the distance of the common dolphin from the coast across different time periods (annually; four-year period; before and after the Sardine Ban)*Alternative Hypothesis (H_1_): *There is a statistically significant difference in the distance of the common dolphin from the coast across different time periods (annually; four-year period; before and after the Sardine Ban)*

It is predicted that common dolphins will be found closer to the coast during years or periods with higher sardine biomass and farther offshore in years following reduced sardine availability. Temporal analyses are expected to reveal shifts in dolphin coastal proximity that align with major fluctuations in sardine biomass, especially around the period of the Sardine Ban’s implementation.**Hypothesis** **2:**The Null Hypothesis (H_0_): *There is no statistically significant relationship between sardine biomass levels and the distance of the common dolphin from the coast.*Alternative Hypothesis (H_1_): *There is a statistically significant relationship between sardine biomass and the distance of the common dolphin from the coast.*

It is predicted that there will be a significant relationship between sardine biomass and the common dolphin’s distance from the coast. It is predicted that in years with lower sardine biomass, common dolphins will occur further from the coast, reflecting reduced nearshore foraging opportunities due to lower prey abundance.

This approach aims to enhance understanding of predator/prey dynamics in marine ecosystems and provide critical information to support evidence-based management and conservation strategies, particularly in the context of fisheries management, dolphin conservation, and MSP.

## 2. Materials and Methods

### 2.1. Study Area

Mainland Portugal is located in the western part of Southern Europe and lies within the ICES division 9a, which is part of the Bay of Biscay and Iberian Coast ecoregion [58]. This study covers the Portuguese mainland coast (Figure 1), extending 860 km from Caminha (41°50′ N, 8°50′ W) to Vila Real St. António (37°12′ N, 7°25′ W). Changes in important topographic, bathymetric, and oceanographic features occur along the Portuguese mainland coast, leading to varying levels of biological productivity. The north–central western coast from Caminha (41°50′ N, 8°50′ W) to Peniche (Cape Carvoeiro) is characterised by long stretches of sandy beaches, a wide and flat continental shelf (40–70 km in width), and strong, fairly homogeneous upwelling with northern wind regimes. These regimes result in colder waters with high productivity. This region also contains one of the largest submarine canyons in Europe, the Nazaré Canyon, located at approximately 39°40′ N, 009°–011° W.

The central–southwestern coast, located between Peniche (Cape Carvoeiro) and the western tip of Cabo S. Vicente, has a narrower (10–20 km wide) continental shelf and is also characterised by strong upwelling. The southern region, also known as the Algarve, extends from the southern tip of Cabo S. Vicente to Vila Real St. Antonio (37°12′ N, 7°25′ W). This region has a very narrow continental shelf (5–20 km wide), warmer waters influenced by the Atlantic and Mediterranean currents [59], and prevailing southerly winds. It is also influenced by several rivers (e.g., Arade, Guadiana) and highly productive coastal lagoons (e.g., Ria Formosa and Ria de Alvor). Furthermore, the entire mainland coast represents a transition between the warmer lower latitudinal waters of the Mediterranean Sea and the cooler higher latitude waters of the northeastern Atlantic [60].

### 2.2. Data Collection

European sardine biomass and common dolphin sighting data were obtained as part of the research surveys led by the Portuguese Institute for the Sea and the Atmosphere (IPMA). These data were collected from systematic vessel surveys from 2005 to 2011 and 2013 to 2020, covering September to June. There was no survey conducted in 2012 due to lack of funding [61]. The primary objective of these ongoing annual surveys is to determine the biomass and spatial distribution of coastal small pelagic fish, particularly the European sardine, in order to make stock assessments and analyse environmental factors that may influence their survival [62].

The surveys were also used to obtain opportunistic data on seabirds, cetaceans, and marine turtles. Cetacean data were collected by one or two trained observers from the Portuguese Society for the Study of Birds (SPEA, BirdLife’s partner in Portugal) using the European Seabird at Sea Methodology (ESAS) [63,64]. All cetaceans within a 300-meter-wide strip transect on one side of the boat were identified, counted, and their behaviour recorded. The counts were assigned to five minutes of transect effort (resulting in an average line length of 1.3 km). These segments serve as the sampling unit considered for further analysis here. A more detailed description of the sampling methodology can be found in [63,64]. Environmental data, such as Beaufort sea state and visibility, were also recorded.

### 2.3. Data Analysis

#### 2.3.1. Temporal Stratification

The encounter rate, the number of animals per unit length, calculated for each 5-min segment, was obtained from the data. For relevant comparisons, the pooled encounter rate was also calculated for different periods, including:(i)Every year;(ii)Every four years;(iii)Before and after the Sardine Ban.

To allow for spatial comparisons before and after the Sardine Ban, two periods were established: 2005–2011 (before the ban) and 2013–2020 (after the ban). Additionally, to refine the temporal analysis, four-year periods were also analysed (2005–2008; 2009–2012; 2013–2016; 2017–2020).

Bar charts were created using R (R Core Team, 2024 from within RStudio Version 4.1.2 [65] using ggplot2 [66] to visualise differences in the encounter rate across the different years.

#### 2.3.2. Sightings and Grid Maps

To visualise common dolphin occurrences, a sighting map was created on QGIS (Version 3.10.0) [67]. As a supplementary means to visualise the data, a heat map of common dolphin occurrences was created using the Kernel Density Estimation tool on QGIS. The Kernel Density Estimation applied a kernel radius of 0.1 × 0.1 km. Kernel Density Estimation is a non-parametric method that can be used to estimate the probability density function, f(x), of a random variable, X. The kernel function used in this study was based on the quartic kernel function.

However, it is important to note some of the potential limitations of this method. Specifically, the smoothing effect in the Kernel Density Estimation can lead to artificial hotspots in areas where there are no sightings. As such, this method was not used as the primary method but rather as a supplementary tool to visualise common dolphin occurrences.

Grid maps were generated to compare the common dolphin encounter rate across different years and seasons, allowing different periods to be visualised on a consistent density scale. The common dolphin encounter rate was quantified in a 10 km^2^ orthogonal grid using QGIS, and the average encounter rate was calculated for each grid cell.

#### 2.3.3. Distance to Coast

Changes in the distance of the common dolphin from the coast were investigated to understand their movements in relation to prey availability. The hypothesis is that the common dolphin will occur significantly further from the coast during periods of low sardine biomass to compensate for low prey availability in inshore waters [58]. The null hypothesis is that there will be no significant difference in the distance of the common dolphin from the coast across the different time periods (i.e., annually, four-year period, before and after the Sardine Ban).

Distance-to-coast values for each common dolphin sighting were obtained by calculating the shortest distance (in metres) to a shapefile of the European coast (https://www.eea.europa.eu/data-and-maps/data/eea-coastline-for-analysis-1/gis-data/europe-coastline-shapefile) (accessed 7 March 2021). The mean, standard deviation, and coefficient of variation (CV) of the distance to the coast were obtained for each time period.

A boxplot was created to visualise differences in the distance of the common dolphin from the coast between each time period. The mean distance of common dolphins from the coast was plotted against the total sardine biomass data from 2005 to 2020 to visualise trends between the two variables.

The Shapiro–Wilk test was used to test for the normality of residuals from a model of common dolphins’ distance to coast per period. Since the residuals significantly deviated from a Gaussian distribution, a non-parametric test was applied to evaluate differences in dolphins’ distance to the coast across the various time periods. A Wilcoxon test was used to test whether there was a significant difference in the distance to the coast before and after the Sardine Ban. A Kruskal–Wallis test was used to test for a statistical difference in the distance to the coast for the different years and seasons. Dunn tests for pairwise comparisons were applied to understand which time periods were statistically different.

In addition, to assess the relationship between sardine biomass and the common dolphins’ distance from the coast, a Generalised Additive Model (GAM) model was fitted to the data using the mgcv package in R [65,68]. This modelling approach allows for flexible, non-linear relationships using a smooth function.

The following equation was provided:log(E[distance]) = β_0_ + s(sardine biomass)
where

E[distance] represents the expected value of distance of common dolphins’ distance from the coast;β_0_ is the intercept value;s(biomass2) is a smooth function (spline) of sardine biomass.

A Gamma distribution was assumed for the response variable, with a log link function. To visualise the fitted relationship, predictions were generated over sardine biomass values, and 95% confidence intervals were calculated based on the standard errors of the fitted values. Model diagnostics and summaries were examined to assess goodness-of-fit and validate the model assumptions.

The results were plotted to illustrate the predicted dolphin distances from the coast as a function of sardine biomass. All statistical analyses were conducted in R (Version 4.1.2) [65], and a significance level of 0.05 was applied to all statistical tests.

## 3. Results

### 3.1. Sighting and Grid Maps

A sighting map of common dolphins from 2005 to 2020 is presented in Figure 1. The common dolphin showed the highest concentration of sightings off the central–western coast of Nazaré (39°38′ N, 9°07′ W) and Figueira da Foz (40°10′ N, 9°01′ W), as well as along the southern Portuguese mainland coast near Sagres (36°44′ N9, 8°58′ W). There were also milder hotspots off the western coast near Matosinhos (41°08′ N, 9°00′ W), Póvoa de Varzim (41°26′ N, 8°55′ W) and Lisboa (38°41′ N, 9°43′ W).

### 3.2. Encounter Rate

The mean encounter rates of the common dolphin from 2005 to 2020 are shown in Figure 2. The figure indicates a generally increasing trend in the common dolphin encounter rate, with a few atypical years. Encounter rates were lowest in 2014 and peaked in 2018, with the second and third highest encounter rates in 2013 and 2008, respectively.

Common dolphin encounter rates were plotted and compared using grid maps before (2005–2011) and after (2013–2020) the Sardine Ban (Figure 3). Before the Sardine Ban, the main hotspots were located off the central coast between Nazaré and Figueira da Foz (from 39°41′ N, 8°57′ W to 40°27′ N, 8°57′ W), the southern coast near Sagres (36°45′ N, 8°50′ W) and the northern coast near Póvoa de Varzim (from 41°05′N, 8°54′ W). After the Sardine Ban, these previously identified hotspots remained critical, but there was also a noticeable increase in common dolphin occurrence off the coast of Lisboa (38°39′ N, 9°35′ W) and along the western Algarve coast (from 36°46′ N, 8°59′ W).

The encounter rate of the common dolphin was compared across the time periods 2005–2008, 2009–2011, 2013–2016, and 2017–2020 (Figure 4). The highest occurrence was observed in the periods 2005–2008 and 2017–2020. In 2005–2008, there was a high density of sightings along the central and northern coast, particularly around the regions of Figueira da Foz and Aveiro. Between 2009 and 2016, there was a decline in the density of the common dolphin along the entire mainland coast of Portugal, with most sightings occurring farther offshore. In 2017–2020, there was a noticeable increase in dolphin density, with hotspots reappearing off the central coast near Nazaré and off the southern coast near Sagres.

### 3.3. Distance from the Coast

The Kruskal–Wallis test indicated statistically significant differences in the distance of the common dolphin from the coast across years (*p*-value 7.41x10^09^). The greatest mean distance to the coast was recorded in 2017 (μ = 32.19 ± 15.43 km), followed by 2014 (μ = 25.93 ± 10.57 km) and 2011 (μ = 56.61 ± 12.03 km).

The Kruskal–Wallis and subsequent pairwise comparison showed that there was a significant increase in the mean distance to the coast across the four-year periods (*p*-value < 0.0003). Significant differences were observed between the following periods:2005–2008 (μ = 20.67 ± 10.86 km) and 2013–2016 (μ = 22.16 ± 12.30 km);2009–2012 (μ = 16.69 ± 10.86 km) and 2013–2016;2013–2016 and 2017–2020 (μ = 19.54 ± 12.30 km).

The highest mean distance from the coast was observed during 2013–2016. However, no statistically significant differences were found in the mean distance to the coast (a) before and after the Sardine Ban (*p*-value = 0.50) and (b) between summer and winter (*p*-value = 0.19).

Figure 5 plots the mean distance of the common dolphin against sardine biomass estimates provided by the Working Group on Southern Horse Mackerel, Anchovy, and Sardine (WGHANSA) for the survey period [69]. The figure suggests an inverse relationship between the total sardine biomass and the common dolphins’ distance from the coast. The years when the common dolphin was located furthest from the coast (2011, 2014, and 2017) coincided with the years when sardine biomass was at its lowest.

The GAM modelled the effect of sardine biomass on the distance of the common dolphin from the coast as a smooth term (Figure 5 and Figure 6). At low sardine biomass levels, dolphins were predicted to occur farthest from shore. As biomass increased to approximately 250,000 tons, dolphins moved progressively closer to the coast. Beyond this point, further increases in sardine biomass were associated with a gradual increase in distance from the shore, although the effect was less pronounced than at lower biomass levels, eventually reaching a plateau.

The estimated degrees of freedom for the smooth term of sardine biomass was 2.87, indicating a potential non-linear effect. However, the relationship between sardine biomass and distance from the coast was not statistically significant (*p* = 0.084; R^2^ = 0.382). The adjusted R^2^ value indicated that 50.1% of the deviance in the data was explained by the model.

While the results suggest a potential association between the sardine biomass and the distance of the common dolphin from the coast, the evidence does not support a statistically significant conclusion. As such, the conclusions should be interpreted with caution.

## 4. Discussion

### 4.1. Overview of the Study

This study represents the first investigation into the habitat use of the common dolphin along the western Iberian coast and provides insight into critical hotspot areas that should be considered in future conservation and management efforts. The research is based on a unique, long-term, systematically collected dataset from vessel-based surveys between 2005 and 2020, aimed at understanding how common dolphin distribution varies spatially and temporally. These surveys, along with the accompanying analysis, offer a valuable understanding of the common dolphin distribution along the western Iberian mainland coast.

The study is intended to support the identification of priority conservation areas, as has been successfully done elsewhere. Until now, there has been limited dedicated research on the distribution of the common dolphin along the western Iberian coast. The lack of knowledge about the common dolphins’ core habitat in the area is particularly concerning given the high-intensity level of anthropogenic activities, such as fisheries and maritime tourism, that affect the area [32,34,55,56,70].

The findings show that between 2005 and 2020, the highest number of sightings occurred off the central–western and southern mainland coast, particularly from Figueira da Foz to Nazaré, and off the windward area of the southern coast, namely Sagres. These common dolphin hotspots are likely related to the presence of geographical and oceanographic features that favour high productivity [71]. For instance, there are high densities of common dolphins located near the major Portuguese submarine canyons (i.e., Nazaré, Lisboa/Setubal, and Cape São Vicente). Submarine canyons act as important channels for the upwelling of nutrients to surface water, therefore promoting high biological productivity and abundance of fish and crustacea [72,73]. Other studies have shown that common dolphin abundance is high off Nazaré and Sesimbra, where the topographic features promote upwelling and high productivity [74]. This aligns with the findings from the current study, which shows a high abundance of common dolphins in these locations.

Furthermore, studies have shown that the common dolphin tends to aggregate near large river mouths due to the high nutrient runoff [75,76]. These river mouths act as critical fish recruitment sites and spawning grounds due to their high productivity [77]. The high number of dolphin sightings in these sites suggests that they might be specifically aggregating on these important recruitment grounds due to increased prey availability.

In this context, the region between the Nazaré Canyon and the Minho River on the western Portuguese shelf is known as a major sardine (*Sardina pilchardus*) spawning ground [78], reinforcing its significance as a key foraging area for common dolphins. There is a high density of dolphins located near the mouth of the Rio Mondego (Figueira da Foz) and a moderate density near other major estuaries and rivers such as the Rio Douro (Porto), Ria de Aveiro (Aveiro), the Rio/Estuary of Tejo (Lisbon), and the Rio/Estuary of Sado (Setúbal). This pattern of coastal aggregation in productive estuarine zones is also reported for other coastal cetaceans, such as bottlenose dolphins (*Tursiops truncatus*) in the Adriatic Sea and in the Gulf of Mexico, where there is increased occurrence near river plumes due to high prey availability (e.g., [79,80]).

### 4.2. Common Dolphin Encounter Rate and Distribution Associated with the Sardine Ban

As previously mentioned, common dolphins tend to aggregate in areas with high prey abundance. Thus, it is expected that changes in prey availability will likely affect their distribution and density [74]. Since 2006, there has been a significant reduction in sardine biomass due to changes in sardine recruitment caused by intensive fishing practices and environmental factors [52,53]. As a result, the Sardine Ban was implemented in 2012, introducing a set of regulatory and technical fishing measures aimed at preventing the further decline of sardine stocks [81].

One of the objectives of this study is to examine the relationship between changes in sardine stock levels, especially due to the Sardine Ban, and the common dolphin occurrence. Indeed, the study indicates that the common dolphin encounter rate followed a similar trend to sardine biomass levels [58]. The grid maps show a low encounter rate of common dolphins close to the coast from the years 2009 to 2016. This low encounter rate coincides with the period when sardine biomass was at its lowest levels around coastal areas [81]. For instance, in 2009, sardine landings exceeded ICES scientific recommended advice by 24% [53]. The combined effects of overfishing, environmental impacts, and trophic interactions led to peak mortality and the lowest recorded level of recruitment for sardine stocks in 2011 [82]. In the current study, there was also a particularly low encounter rate of the common dolphin in 2011, which suggests reduced prey availability prior to the implementation of the Sardine Ban. This aligns with similar studies for other cetacean species in the Mediterranean area and the Pacific region [77,83].

After the implementation of the Sardine Ban in 2012, there was a gradual recovery in sardine stock levels along the Portuguese mainland coast [84] up to 2020. Aligning with this gradual increase in sardine stock biomass, there is an increase in the common dolphin encounter rate and density from 2016 onward. This may suggest that the dolphins could be benefiting from the sardine stock recovery and responded positively to the implementation of the Sardine Ban.

There are inconsistencies between common dolphin encounter rates and sardine biomass in 2008, 2013, and 2018, which represent years with unusually high encounter rates. While some interannual variability in the months sampled was observed, the sampling scheme during these years did not differ significantly from other years, supporting the robustness of the observed patterns and reducing the likelihood of sampling bias as an explanation for the peaks. The implementation of the Sardine Ban may have contributed to an increase in sardine biomass, which could partially explain the higher encounter rates observed in 2013 and 2018. Behavioural factors, such as increased social aggregation or reproductive activities, could have further influenced their distribution and detectability. Additionally, these anomalies may be explained by the complexity of other factors (e.g., behaviour, environmental) influencing dolphin distribution along the Portuguese mainland coast. These findings underscore the need for further research integrating prey distribution data, oceanographic conditions, and dolphin behaviour to better understand the drivers of these interannual variations.

In addition to temporal patterns in encounter rate, the study revealed spatial shifts in the common dolphin distribution. The dolphins were recorded significantly further from the coast between 2013 and 2016, which represents a period of historically low sardine biomass. This shift could suggest that dolphins may have expanded their foraging range offshore in response to the reduced availability of their preferred prey in coastal waters when the sardine fishery was subject to significant restriction [85,86]. Distributional shifts of small and large cetaceans have been documented globally, demonstrating their ecological plasticity in response to changes in prey abundance [87]. Such shifts are also observed in other marine top predators like northern gannets, which have been shown to adjust their foraging range based on prey distribution [88].

Previous studies have described common dolphins as opportunistic feeders known for their dietary flexibility [33]. The dolphins may have exploited localised prey aggregations, including alternative prey species, during years with low inshore sardine abundance. While sardines remained the prevalent component in the diet based on stomach content analyses, other small pelagic fish species, such as Atlantic chub mackerel (*Scomber colias*) and horse mackerel (*Trachurus trachurus*), constituted a greater proportion of the common dolphins’ diet than in previous years [33,37]. This adjustment likely reflects opportunistic foraging in areas where sardines were scarce or less accessible to purse seine fleets.

In 2017, as sardine stocks began to recover and fishing restrictions were relaxed, common dolphins were observed closer to the coast. Dias et al. (2022) [35] showed that there is a higher probability of occurrence of dolphin/fishery interactions in areas with higher fishing effort. Thus, dolphins may be moving closer to the coast to benefit from the higher sardine availability inshore and the prey aggregation favoured by the purse seine fishery [32,35]. Similar interactions between small cetaceans and fishing fleets have been observed in other regions, such as the Mediterranean and South Australia, where increased prey aggregations near purse seiners or bottom trawlers attract dolphins (e.g., [89,90]).

When modelled, biomass did not have a significant effect on common dolphin distance from the coast. This suggests that the spatial distribution of dolphins is likely influenced by additional environmental or ecological factors not accounted for in this study, such as sea surface temperature, oceanographic currents, or seasonal shifts in the availability and distribution of other prey species. This is consistent with global research showing that climatic factors influence the habitat use and migratory behaviour of cetaceans [91].

### 4.3. Limitations and Future Directions

Abundance estimates are essential for assessing the conservation status of a population and the impact of anthropogenic threats (EU Habitat Directive 92/43/EEC; [92,93]). This study considered a simpler framework. Rather than estimating dolphin density, the study considers encounter rates, which, by definition, are indexes of relative abundance and are not corrected for detectability. Although there is no strong reason to suggest that there would be a relevant correlation between detectability and distribution to obscure real effects or create spurious ones, this remains an untested assumption worth exploring.

While estimating detectability may be feasible within a distance sampling context, this was beyond the objectives of the current work, where our primary aim was to explore course scale spatial and temporal patterns in relation to the sardine fishery. Additionally, besides including detectability, a distance sampling approach could be extended to models of spatial density over space, relating the dolphin density to other biotic and abiotic covariates. It is likely that there are environmental variables beyond prey availability that are influencing the movements and distribution of the common dolphin. For instance, environmental variables such as water depth [94], surface temperature [95], and seabed gradient [96]. Future studies should incorporate these environmental variables to better understand their influence on the dolphins’ distribution. This methodology could be particularly useful for predicting the distribution of the common dolphin, given their widespread distribution [97]. Models such as Density Surface Modelling can be used to locate potential hotspots in situations where sampling coverage is limited, therefore providing valuable information for decision-makers and managers [98]. Such modelling techniques would contribute to a better understanding of temporal and spatial changes in cetacean abundance, therefore allowing managers to make more informed conservation decisions.

## 5. Conclusions

The study investigates the distribution of the common dolphin, the most frequently observed cetacean species, along the western Iberian coast. The study is the first to use long-term, standardised survey data to assess common dolphin distribution over a 16-year period. The findings reveal that common dolphins showed response movements linked to fluctuations in sardine stock levels, their primary prey. During years when sardine levels were at their lowest, dolphin densities declined noticeably closer to the coast, where sardines were once abundant. After the implementation of the Sardine Ban in 2012, there was a gradual increase in the sardine stock levels, which resulted in the dolphins moving closer to shore.

These findings suggest that common dolphin habitat use is closely tied to prey availability, with fluctuations in prey stocks potentially influencing their spatial distribution. The identification of common dolphin hotspots along the western Iberian coast provides valuable information in MSP, Marine Protected Area (MPA) designation, and the development of effective management strategies. The results offer critical insights for managers and decision-makers to identify high-risk areas for common dolphins during the MSP process. Conservation management measures should be continuously implemented and enforced to mitigate the negative anthropogenic impacts on common dolphin populations, which is essential for ensuring a healthy marine ecosystem.

## Figures and Tables

**Figure 1 animals-15-01552-f001:**
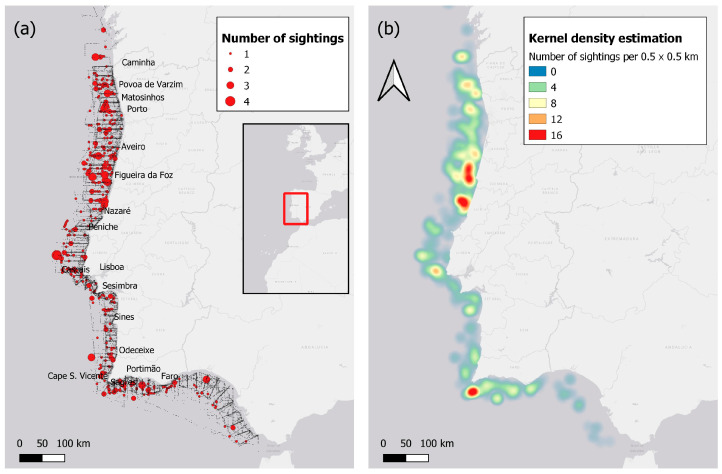
Common dolphin sightings from 2005 to 2020: (**a**) Number of sightings during transect surveys (grey dots denote the transect); (**b**) Heat map of sightings per 0.5 km × 0.5 km using kernel density estimation.

**Figure 2 animals-15-01552-f002:**
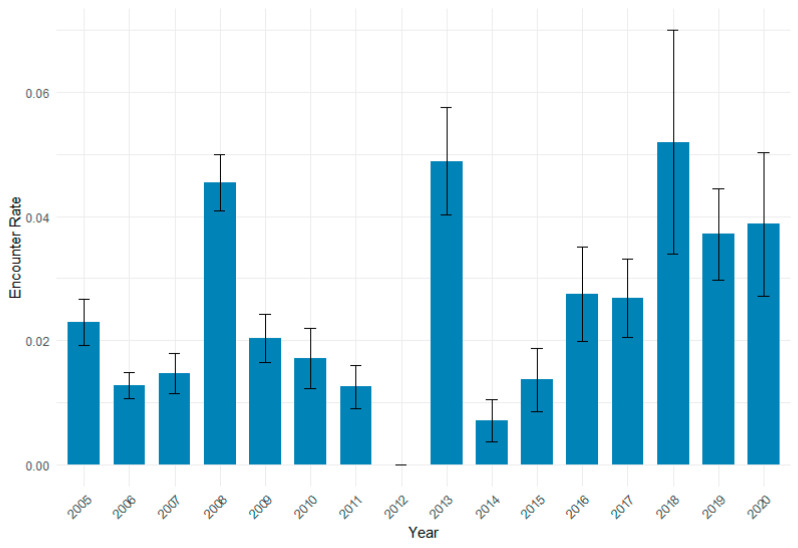
Mean encounter rate of the common dolphin per year (with standard error) across segments.

**Figure 3 animals-15-01552-f003:**
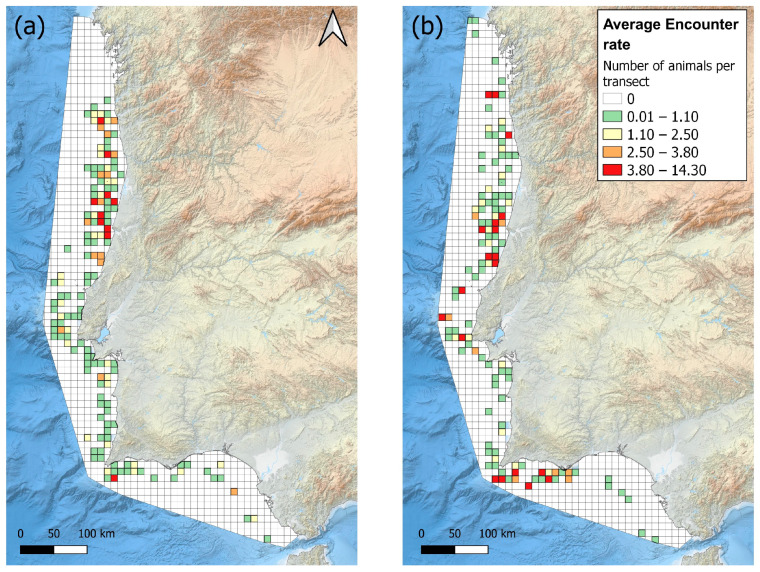
The encounter rate (animals/km travelled) of the common dolphin (**a**) before and (**b**) after the Sardine Ban of 2012. Grid map with a cell resolution of 10 km^2^.

**Figure 4 animals-15-01552-f004:**
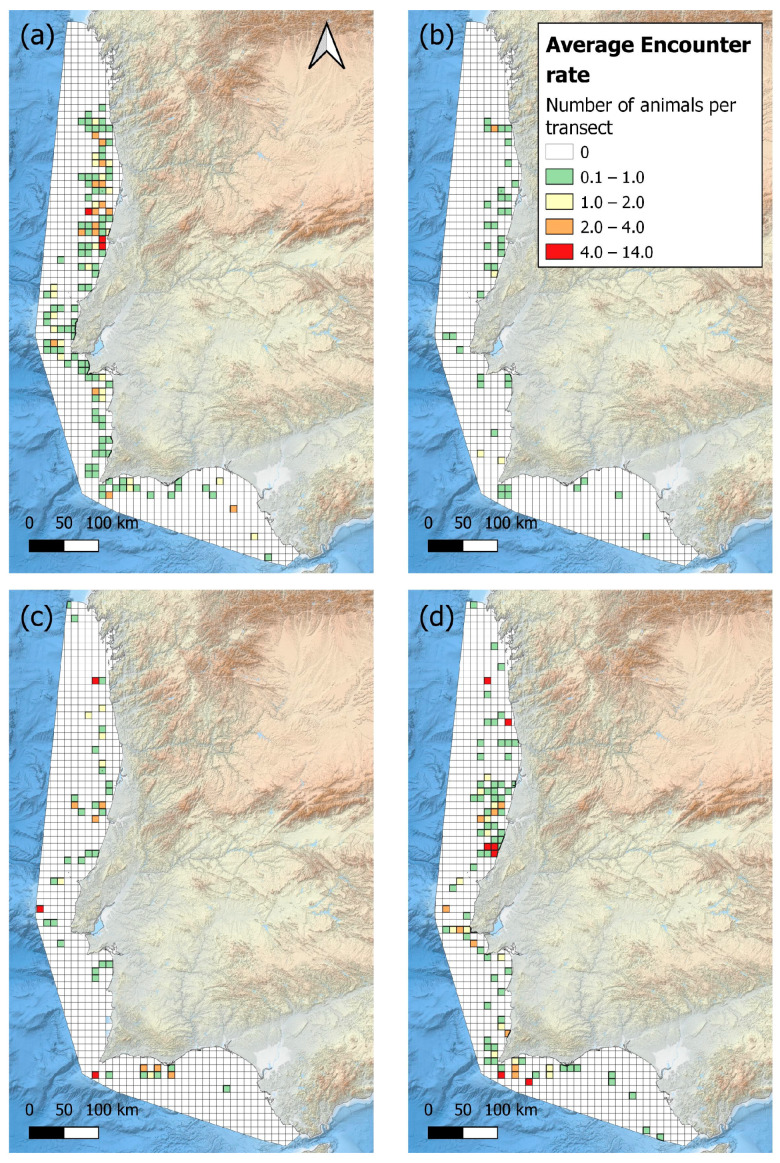
The encounter rate (animals/km travelled) of the common dolphin for every four years: (**a**) 2005–2008, (**b**) 2009–2011, (**c**) 2013–2016, and (**d**) 2017–2020. Grid map with a cell resolution of 10 km^2^.

**Figure 5 animals-15-01552-f005:**
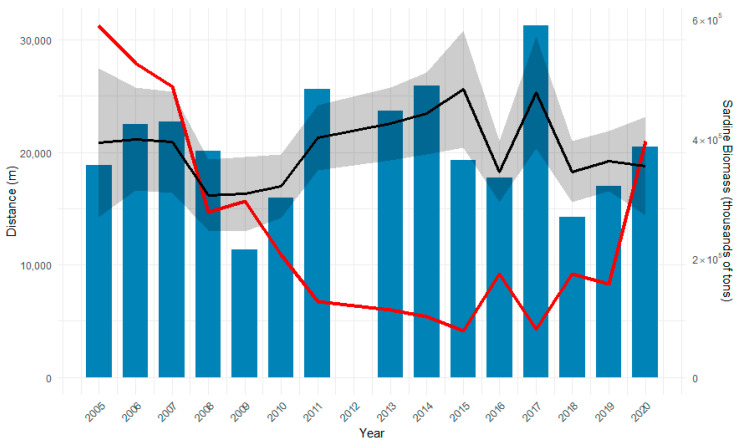
Total sardine biomass (tons, red thick line) plotted with the mean distance from the coast of the common dolphin for each year (blue box plot). Black line represents the predicted dolphin distance from the coast based on sardine biomass, using a smooth GAM curve with a 95% confidence band (grey shaded area); sardine biomass data taken from WGHANSA [58].

**Figure 6 animals-15-01552-f006:**
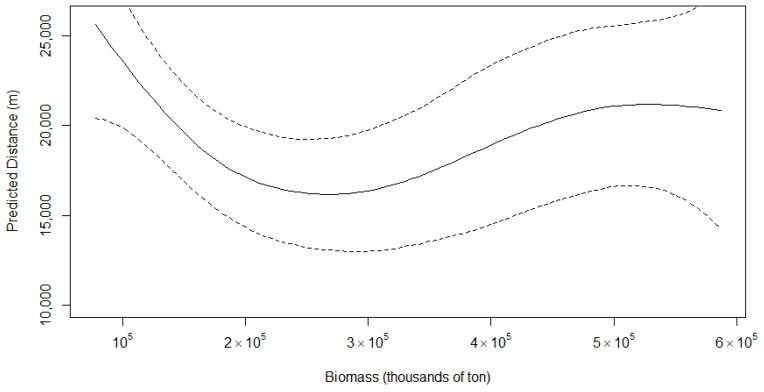
Predicted distance (m) of the common dolphin from the coast based on a GAM, with sardine biomass (thousands of tons) as the predictor (dashed line represents corresponding 95% confidence bands).

## Data Availability

The datasets generated and/or analysed during the current study are available from the corresponding author upon request.

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
