# Peer review of "When Sardines Disappear: Tracking Common Dolphin, Delphinus delphis, Distribution Responses Along the Western Iberian Coast"

_animals, 2025, doi:10.3390/ani15111552_

Round 1
Reviewer 1 Report
Comments and Suggestions for Authors
In the paper “When sardines disappear: Tracking Common Dolphin, Delphinus delphis, distribution responses along the Western Iberian Coast” the authors provides the first insights into common dolphin distribution along the Portuguese mainland coast, using sighting data from vessel research surveys (2005-2020) to identify hotspot areas, accounting for monthly and seasonal distributions overlapping with sardine abundance. This manuscript is well organized, and the drawn conclusions are coherent with the obtained results. I hope to provide very useful suggestions to improve the overall clarity of your study as well as the quality of your analysis. I think that my suggestions look feasible to you, and I believe you will be able to address them.
Lines 47 – 49: Please arrange the keywords alphabetically.
Lines 54 – 55: I think that you should add these two important references as examples to support your sentence: “In marine and coastal management, the use of Species Distribution Models (SDM) can be used to inform Marine Spatial Planning (MSP) initiatives,”. I would like to suggest:
Bosso, L., Panzuto, R., Balestrieri, R., Smeraldo, S., Chiusano, M. L., Raffini, F., ... & Gili, C. (2024). Integrating citizen science and spatial ecology to inform management and conservation of the Italian seahorses. Ecological Informatics, 79, 102402.
Karp, M. A., Cimino, M., Craig, J. K., Crear, D. P., Haak, C., Hazen, E. L., ... & Woodworth-Jefcoats, P. A. (2025). Applications of species distribution modeling and future needs to support marine resource management. ICES Journal of Marine Science, 82(3), fsaf024.
Lines 55 – 57: I think that you should add these two important references as examples to support your sentence: “…by providing insight into potential areas of conflict between different users of the maritime space.”. I would like to suggest:
Bosso, L., Raffini, F., Ambrosino, L., Panzuto, R., Gili, C., Chiusano, M. L., & Miralto, M. (2025). Geoportals in marine spatial planning: state of the art and future perspectives. Ocean & Coastal Management, 266, 107688.
Galparsoro, I., Montero, N., Mandiola, G., Menchaca, I., Borja, Á., Flannery, W., ... & Stelzenmüller, V. (2025). Assessment tool addresses implementation challenges of ecosystem-based management principles in marine spatial planning processes. Communications Earth & Environment, 6(1), 55.
Lines 64 – 66: I think that you should add these two important references as examples to support your sentence: “Cetaceans are considered important umbrella species due to their ecological significance and the broad protection their conservation affords to other marine species and habitats.”. I would like to suggest:
Russo, D., Sgammato, R., & Bosso, L. (2016). First sighting of the humpback whale Megaptera novaeangliae in the Tyrrhenian Sea and a mini-review of Mediterranean records. Hystrix, 27(2), 219.
Di Sciara, G. N., Hoyt, E., Reeves, R., Ardron, J., Marsh, H., Vongraven, D., & Barr, B. (2016). Place‐based approaches to marine mammal conservation. Aquatic conservation: marine and freshwater ecosystems, 26, 85-100.
Lines 115 – 123: Please explain in detail your hypothesis and predictions. You need to expand this section if you would want to express exactly what you want to do.
Lines 148 – 165: Did you analyse your spatial data for potential autocorrelation?
Line 177: What is the package? Please note that all the R codes must be added in the supplementary materials well commented.
Lines 219 – 229: Please provide more details on you modelling procedure.
Lines 323 – 469: The paper discussed not appropriately the context and the theme. In fact, there is important literature not cited by the authors. I think that the authors should discuss their results also comparing them with those already published on other species/genus/family of cetaceans and marine organisms.
Comments on the Quality of English LanguageThe English could be improved to more clearly express the research.
Author Response
We appreciated the constructive comments and suggestions from Reviewer 1, which we have incorporated into the manuscript. Below, in the attached word document, you can find a point-by-point response to all the comments. In bold you can find the respective answer to the comment or question previously stated; referred lines with corrections are linked to the “clean version”. In the online system you can also find:
- a revision with tracked changes
- a revision with all changes accepted ('clean' file)

Reviewer 2 Report
Comments and Suggestions for Authors
- This paper investigates an ecologically significant issue by analyzing the relationship between sardine biomass fluctuations along the Portuguese coast and the distribution patterns of common dolphins. Moreover, this study analyzed 16 years of data, providing substantial ecological value for the research results. However, several areas need revision to reach the acceptance level.
- Some figures lack units, scales, or consistent resolution (e.g., Figures 3 and 4 on pages 8 and 9). We recommend that the authors consider adding the missing legends and scales to ensure visual clarity of the hotspot areas—consistent labeling.
- Page 10, line 314: Although p = 0.084 (not significant), the interpretation of the GAM model results is relatively positive. This weakens the association between sardine biomass and dolphin distance from the coast. It is suggested that the author reframe the result with caution. More specifically, acknowledge that the association is only suggestive rather than a statistical conclusion.
- Although the references are back up the statement, the discussion section sometimes lacks an exploration of the latest or global research on dolphin spatial ecology. It is suggested that the author could consider citing recent global meta-analyses or reviews, such as studies on climate or fishing-induced changes in the distribution of marine predators, etc..
- For the manuscript, it is recommend that authors make minor but thorough language revisions. It is recommended that manuscripts use consistent tenses. For example, avoid mixing present and past tenses when describing results.
Author Response
We appreciated the constructive comments and suggestions from Reviewer 2, which we have incorporated into the manuscript. Below, in the attached word document, you can find a point-by-point response to all the comments. In bold you can find the respective answer to the comment or question previously stated; referred lines with corrections are linked to the “clean version”. In the online system you can also find:
- a revision with tracked changes
- a revision with all changes accepted ('clean' file)

Reviewer 3 Report
Comments and Suggestions for Authors
This manuscript presents a long-term analysis (2005–2020) of the spatial and temporal distribution of Delphinus delphis along the Portuguese coast in relation to sardine (Sardina pilchardus). The study fills a significant gap in ecological research and provides relevant insights for marine spatial planning and fisheries management. The authors analyse an extensive dataset spanning 16 years, offering temporal coverage of common dolphin sightings in Western Iberia. The study supports predator-prey theory and illustrates the utility of cetaceans as ecosystem indicators in the context of changing prey availability and human pressures. This manuscript offers a valuable contribution to cetacean ecology and fisheries management in the Northeast Atlantic. The findings are novel, relevant, and well contextualized.
The manuscript is suitable for publication after minor revisions listed below.
Dolphin data were collected incidentally during fish stock surveys, possibly introducing spatial or temporal bias. However, GAM model fails to reach statistical significance (p = 0.084) even though GAM visually supports a relationship between sardine biomass and dolphin offshore movement. I recommend for the authors to incorporate additional covariates known to influence cetacean distribution (e.g., temperature, depth, salinity) to improve model explanatory power as mentioned in lines 397-398 and 460-463. Why the authors did not incorporate such variables in the analysis? This would integrate environmental component with biotic factors, such as dolphin distribution, an issue that has been already mentioned at the introduction (lines 91-92) as well as the high nutrient runoff discharged from near large river mouths (lines 351-352).
Author Response
We appreciated the constructive comments and suggestions from Reviewer 3, which we have incorporated into the manuscript. Below, in the attached word document, you can find a point-by-point response to all the comments. In bold you can find the respective answer to the comment or question previously stated; referred lines with corrections are linked to the “clean version”. In the online system you can also find:
- a revision with tracked changes
- a revision with all changes accepted ('clean' file)
